# Candidate Key Proteins of Tinnitus in the Auditory and Motor Systems of the Thalamus

**DOI:** 10.3390/ijms26125804

**Published:** 2025-06-17

**Authors:** Johann Gross, Marlies Knipper, Birgit Mazurek

**Affiliations:** 1Tinnitus Center, Charité-Universitätsmedizin Berlin, 10117 Berlin, Germany; birgit.mazurek@charite.de; 2Leibniz Society of Sciences Berlin, 10117 Berlin, Germany; marlies.knipper@uni-tuebingen.de; 3Department of Otolaryngology, Head and Neck Surgery, Tübingen Hearing Research Center (THRC), Molecular Physiology of Hearing, University of Tübingen, 72076 Tübingen, Germany

**Keywords:** auditory perception, biomarker, synaptic transmission, thalamus, tinnitus

## Abstract

To determine candidate key proteins involved in synaptic transmission in the thalamus in tinnitus, we used bioinformatic methods by analyzing protein–protein interaction networks under different conditions of acoustic activity. The motor system was used to analyze the specificity of the response reaction in the auditory system. The databases GeneCard, STRING-, DAVID-, and Cytoscape version 3.9.1 were applied to identify the top three high-degree proteins, their high-score interaction proteins and the gene ontology—biological processes (GO-BPs) associated in the thalamus with synaptic transmission in tinnitus. Under normal hearing conditions, a balanced state of functional connectivity was observed for both systems, the auditory system and the motor system of the thalamus. Under conditions of acoustic stimulation, the GO-BP-enrichment analyses suggest that in the auditory system, tinnitus-related proteins may be involved in responses typically associated with “xenobiotic stimuli”; in the motor system, the activation of the dopaminergic system was observed. Under conditions of tinnitus in the auditory system, key proteins and the GO-BPs indicate the regulation of different developmental processes and regulation by microRNA transcription; in the motor system, tinnitus is also identified as “xenobiotic” but responded with GO-BPs, corresponding to various signaling systems, e.g., tachykinin. Key proteins and their interactions with neurotransmitter receptors may be useful indicators for tinnitus-associated changes in synaptic transmission in the thalamic auditory system.

## 1. Introduction

Tinnitus is the subjective perception of sounds when no external acoustic stimuli are objectively present. It affects about 10–15% of the adult population. Depending on the severity, tinnitus can be a very large psychological and disease burden on the afflicted individual and can result in enormous health care costs [1,2,3]. During the hearing process, sound waves are converted into neural signals in the cochlea, where the frequency and level of the sound are coded. Synaptic transmission is a fundamental process in normal hearing and in tinnitus, referring to the communication between neurons through chemical or electrical signals at synapses. Numerous neuronal centers in the CNS are involved in the processing and perception of sound. Important centers for the processing of acoustic signals from the hair cells are the spiral ganglion (SG), located in the cochlea; the cochlear nucleus (CN), the inferior colliculus (IC), and the thalamus, located in the midbrain; and the cortex, where sound is perceived. In each auditory nucleus, the signals are detected, processed, and transmitted to other centers [4]. The pathophysiological changes in tinnitus are different in each center and range from changes in synaptic transmission and altered protein–protein interactions to cell death and cell regeneration [5].

In previous work, we have characterized key proteins in the protein–protein interaction networks (PPI) in the SG [6], the CN [7], and the IC [8]. We observed that proteins with high degrees (HDPs) and the corresponding high-score interaction proteins (HSIPs; high combined score), together named key proteins, may play a functionally important role in the regulation of protein–protein interactions in acoustic signaling. In all three regions, BDNF is the top one protein with the most interactions with other proteins in tinnitus (Tin). There are clear differences in the top two high-degree proteins and their interaction partners: in the SG, it is NGF, with close interactions with NGFR and NTRK1; in the CN, it is FOS, with high-score interactions with JUN, CREB1, EGR1, MAPK1, and MAPK3; and in the IC, it is GFAP, with high-score interactions with S100B. The analysis indicates that in the SG of Tin, remodeling occurs at the cellular level under the influence of NGF and NGFR, accompanied by both cell death and the generation of new cells. In the CN, the focus is on the modulation of synaptic transmission, influenced by NTF3, NTRK1, and NTRK3 as well as a large number of transcription factors. In the IC, the modulation of synaptic transmission occurs under the influence of NTF3 and NTRK3 and also at the transcription level, but with the involvement of glial cells.

The aim of this study was to identify candidate key proteins in tinnitus in the thalamus, a central structure in the diencephalon of the brain. The thalamus receives abundant sensory information (such as on hearing, vision, touch, and pain) and forwards it to the corresponding areas of the cerebral cortex. The thalamus filters out unimportant stimuli so that the brain can concentrate on essential information [9]. From an anatomical point of view, the thalamus has a number of distinct nuclei, for example, the corpus geniculatum mediale for auditory stimuli and corpus geniculatum laterale for visual information and for motor reactions. The thalamus is part of the so-called ‘cortico-striato-thalamo-cortical loops’ [10]. These circuits enable communication between the cerebral cortex, basal ganglia, and thalamus and are crucial for motor learning and the planning and execution of movements.

Due to the special role of the thalamus as a relay station between the IC, a center that receives auditory signals via the CN, and the cortex, the actual center of perception of tinnitus, the thalamus is attributed a fundamental role in the development of tinnitus [11,12,13,14]. The so-called ‘thalamo-cortical dysrhythmia hypothesis’ of tinnitus is based on human imaging studies [11], the detection of altered cortico-limbic circuitry and its connection to the auditory thalamus and the medial geniculate nucleus (MGN), and the functional connectivity between the MGN and auditory cortices, e.g., in the modulation of temporal characteristics of input signals [12,13]. Neurons in the MGN receive inhibitory and excitatory input from the IC [15]. The top-down projections from the cortex to the MGN do not occur directly but indirectly via basal forebrain projections to the thalamic reticular nucleus [16].

The thalamus plays a central role in the brain’s motor system by acting as an interface between different brain regions. It receives signals from the basal ganglia and the cerebellum, both of which are essential for movement coordination. This information is then transmitted to the motor and pre-motor cortex areas [17]. To verify that the key proteins and the GO terms identified for the “auditory system” of the thalamus are specific for acoustic activation in those areas, in the present work, we compared key proteins and GO terms of the “auditory system” of the thalamus with the key proteins and GO terms of the “motor system” of the thalamus under the conditions of normal hearing (NH), acoustic stimulation (AS), and tinnitus (Tin).

The biological processes involved in tinnitus are made possible by the precisely regulated interaction of numerous proteins, described as the protein–protein interaction (PPI) network. The aim of the present study was to identify candidate key proteins in the thalamus, an important relay station. The analysis of the networks of interacting molecules may contribute to current and future directions in the understanding of the network biology of diseases [18]. The identification of candidate key proteins can help in the discovery of new biomarkers and of new targeted therapies of tinnitus and in the transition to personalized medicine [19,20,21,22,23].

## 2. Results

### 2.1. Auditory System

#### 2.1.1. Gene Sets, Networks, and Key Proteins

Due to different numbers of studies so far available for the different conditions, the gene lists for normal hearing (NH), for acoustic stimulation (AS), and for tinnitus (Tin) in the auditory system (AuS) of the thalamus differ in the number of genes. The most studies are available for tinnitus (Appendix A, Table A1, Table A2 and Table A3). The Venn diagram indicates that there is an overlap of genes between the biological processes analyzed—NH, AS, and Tin (Figure 1).

The protein–protein interaction (PPI) networks of NH, AS, and Tin processes differ in size and structure. The total numbers of nodes are 33 for NH, 35 for AS, and 54 for Tin; the total number of edges are 85 for NH, 201 for AS, and 410 for Tin (Figure 2A–C). The average numbers of neighbors (numbers of interactions between the proteins) is higher in the AS and the Tin networks (NH, 5.4; AS, 11.5; Tin, 15.2). The characteristic path length (shortest distance between any two nodes in the network) of the AS and the Tin networks is accordingly smaller (NH, 2.3; AS, 1.9; Tin, 1.9). The clustering coefficient (the tendency of nodes to cluster together) and the network density (the number of interactions to the number of possible interactions) of the AS and Tin networks are slightly larger than that of the NH network (clustering coefficient NH, 0.44; AS, 0.60; Tin, 0.62; network density NH, 0.18; AS, 0.34; Tin, 0.29) [24,25,26]. To exclude the possibility that these differences are not due to the higher number of proteins in the network, we established these parameters in a tinnitus network of comparable size, indicating that the differences are biologically determined. The key proteins in the network are labeled (see legends).

To define a useful criterion for identifying HSIPs of the top 1–3 HDPs, we first analyzed the frequency distribution of the degree and CS values (Figure 3). The highest frequency was observed at low degree and CS values. The distribution of the degree values has an uncharacteristic form; due to the small number of proteins, the differences between NH, AS, and Tin are not relevant. The distribution of the combined score (CS) values shows high frequencies in the range of low CS scores, a minimum in the range of CS = 800–900, and significantly higher frequencies in the range of CS > 900.

As the key proteins, we selected the top three HDPs and their HSIPs using CS value >90th percentile as the indicator (NH > 937; AS > 904; Tin > 935; Table 1); the 90th percentile cutoff was chosen to limit the number of HSIPs. In all three processes, BDNF is the top one HDP; the top two HDP in the NH process is PVALB and in the AS and Tin processes GFAP. The top three HDP in the NH process is SNAP25, in the AS process is SYP, and in the Tin process is TNF.

#### 2.1.2. GO Enrichment Analysis

The DAVID database allows for the assignment of lists of genes or proteins to the enrichment of gene ontology terms for cellular structures (GO-CCs) and biological processes (GO-BPs). To compare or combine the informative values of the GO terms of the complete gene list with that of the list of key proteins, we determined the GO-CC and GO-BP terms for both lists. To limit the scope of the study, we have restricted the GO terms to five.

The GO-CC terms for NH, AS, and Tin processes that were derived from the complete gene lists and the key protein list are shown in Table 2. The GO-CC terms correspond to well-known synaptic structures and are similar for some terms.

The GO-BPs for the processes NH, AS, and Tin, which can be derived from the lists of genes and the list of key proteins, are shown in Figure 4A–C. The GO termini that are detectable in the NH process (list of genes) correspond to two main processes: (a) perception of sound, which includes synaptic transmission via activation of cholinergic receptors, and calcium ion channels, and (b) the negative regulation of neuronal apoptotic processes to ensure cell viability. Four to eight proteins are involved in these processes; it is interesting to note that key proteins are only involved in the process of the negative regulation of neuronal apoptosis. The BPs derived from the list of key proteins specify these processes and indicate the roles of key proteins BDNF and NTRK2.

In acoustic stimulation (AS, Figure 4B), the GO termini in the list of genes indicate the process of stimulation (expressed in the term “xenobiotic stimulus”) and associated responses (response to hypoxia and activity), the negative regulation of neuron apoptosis, and the positive regulation of transcription via microRNA. Key proteins are involved in biological processes associated with xenobiotic stimulus and apoptosis. The GO-BP termini, which are derived from the list of key proteins, specify these processes, indicating the important role of BDNF and NTRK2, although the statistical significance in the Benjamini–Hochberg test was not achieved. The appearance of “circadian rhythm” suggests that the response to “xenobiotic stimulus” may be dependent on the circadian rhythm.

In the Tin process (Figure 4C), the list of genes includes the biological functions of the regulation of miRNA transcription, of the neuron apoptotic process, and of the peptidyl-serine phosphorylation. It is interesting to note that tinnitus is evaluated as the “sensory perception of sound” together with the “xenobiotic stimulus”. The GO terms of the key protein list specify the functions by indicating the role of cell-surface receptor protein tyrosine kinase, the role of vascular-endothelial growth factor, and the regulation of cell proliferation. It should be noted that in nearly all GO terms of the gene list, several key proteins appear (printed in bold).

#### 2.1.3. Interactions of Key Proteins with Neurotransmitter Receptors

To analyze the effects of key proteins on the synaptic transmission, we examined the interactions of key proteins with the corresponding receptor proteins for the processes NH, AS, and Tin (Figure 5A–C). We found clear differences in the occurrence of the receptor proteins as well as the interactions with key proteins. Two types of neurotransmitter receptors are detectable in the AuS list: glutamate and acetylcholine receptors.

In NH, GRIA3 shows interactions with NTRK2 and SNAP25 (Figure 5A). GRIA3, a subunit of the AMPA receptor, mediates rapid excitatory transmission in the CNS. NTRK2 and SNAP25 influence GRIA activity via different mechanisms: NTRK2 modulates synaptic strength and receptor trafficking, and SNAP25 (synaptosomal-associated protein 25) is involved in synaptic-vesicle exocytosis and thus neurotransmitter release [27]. CHRNA9 and CHRNA10 (subunits of nicotinic acetylcholine receptors) are present in the gene list and are involved in ignoring auditory distraction during visual stimuli [28], without, however, interactions with key proteins.

In the AS process, GRIN1 (Glutamate Ionotropic Receptor NMDA Type Subunit 1) and CHRNA4 (Cholinergic Receptor Nicotinic Alpha 4 Subunit) are detectable (Figure 5B). NMDA receptors mediate excitatory transmission and are involved in the regulation of synaptic plasticity, learning, and memory. Acetylcholine receptors involving CHRNA4 influence cognitive processes and modulate neurotransmitter release and synaptic plasticity. Several key proteins (CREB1, BDNF, NTRK2 SYP, GFAP) show close interactions with GRIN1, so it can be assumed that in particular, GRIN1 is regulated during acoustic stimulation [29,30,31].

GRIN1 is the only neurotransmitter receptor in the Tin gene list; it shows close relationships to BDNF, GFAP, and CREB1 (Figure 5C). The interaction of these proteins in the thalamus is essential for synaptic transmission and plasticity in the thalamus and in particular for the perception of the thalamocortical somatosensory pattern and sensorimotor behavior [32]. Of particular interest is the presence of PDYN (prodynorphin) in the gene list and its close interaction with GRIN1. PDYN is a precursor protein for dynorphins; dynorphins are opioid peptides that play a role in modulating the perception of pain and stress. Dynorphins modulate excitatory signals and are important for adaptation to environmental stimuli. They are involved in the perception of acoustic signals and may contribute to the development of tinnitus by modulating synaptic transmission [33].

### 2.2. Motor System

#### 2.2.1. Gene Sets, Networks, and Key Proteins

To compare the response of the motor system (MoS) of the thalamus to the acoustic signals of NH and AS and to Tin in this system of the thalamus, we determined the same parameters as in the AuS (Appendix A, Table A4–A6). Only a small overlap is observed between the MoS and the AuS in the NH process (Figure 6, 8.8% overlap). The same applies to the processes AS and Tin. In contrast, the overlap between NH, AS, and Tin processes in the MoS of the thalamus is larger (34%) and similar to the AuS of the thalamus region.

Similar to the auditory system of the thalamus, there are clear differences in the structure and topological criteria of the PPI networks between NH, AS, and Tin processes (Figure 7A–C). Compared to the NH network, the average number of neighbors of the AS and Tin networks is larger (NH—4.5, AS—11.0, Tin—8.7), the characteristic path length is slightly smaller (NH—2.2, AS—1.8, Tin—2.0), and the clustering coefficient (NH—0.39, AS—0.63, Tin—0.59) and the network density (NH—0.19, AS—0.38, Tin—0.27) are larger.

The frequency distributions of the degree and CS values (Figure 8) correspond largely to those of the auditory system (Figure 3), with two differences: (a) 26% of the CS values of the NH process are below 400 (in group1), as a possible expression of resting activities, and (b) the overall frequency of high CS values is less pronounced than in the AuS, obviously an expression of a more homogenous activity system under the selected conditions.

The key proteins identified in the motor system (Table 3) differ significantly from those in the auditory system of the thalamus, particularly in the top two and top three HDPs and their HSIPs (Table 1). In the NH process, the top one HDP is AKT1; in the AS and Tin processes, BDNF appears as the top one HDP.

#### 2.2.2. GO Enrichment Analysis

The GO enrichment terms based on the ‘motor system gene lists’ and the corresponding key protein lists differ between the NH, AS, and Tin processes (Figure 9A–C). In the NH process, two types of GO-BPs can be identified by the gene list: (a) Gene expression and phosphorylation processes indicate activity and signal transmission in the CNS; phosphorylation plays an important role in normal metabolism in the regulation of apoptosis, transcription, and gene expression. (b) The term “response to heat” may be regarded as a non-specific term. The key protein list specifies the gene expression as “negative” to limit the process to normal physiological activity. The GO term “positive regulation of glycogen biosynthetic process” refers to the formation of glycogen, a storage form of glucose that serves as an energy source. That can be understood as an expression of the organism’s normal metabolic activity. It is noteworthy that the key proteins BDNF and NTRK2, which play a key role in the auditory system of the thalamus, do not appear in the top GO-BPs in the MoS. The term “peripheral nervous system myelin maintenance” may also be regarded as a non-specific term. In summary, all terms stand for normal physiological processes by which an organism reacts to certain non-specific environmental stimuli to maintain its homeostasis.

In the AS process (Figure 9B), the following GO-BPs are present: (a) The term “response to xenobiotic stimulus” is perceived as an apparently “foreign” signal, in response to which changes in blood pressure and memory occur. (b) These processes are associated with dopaminergic synaptic transmission, which includes the dopamine synthesis process and dopamine uptake. In the motor system of the thalamus, acoustic stimulation induces processes of synaptic transmission, which are a precondition for the successful motor response of the organism to an unexpected danger or signal.

In the tinnitus process, several GO-BP termini appear, which may be related to each other (Figure 9C): (a) The terms “response to xenobiotic stimulus”, to cocaine, to fungi, and to cadmium may be registered in the motor system as a “foreign” signal that leads to changes in synaptic transmission or to oxidative stress. The effects of cocaine and cadmium show numerous interactions, which can also lead to neuroadaptive changes and even neurodegeneration [34]. (b) “Chemical synaptic transmission”, “tachykinin receptor signaling pathway”, and phosphorylation signaling processes could be related to the activation of several pathways, including neuropeptide receptors [35,36,37]. Tachykinin receptors belong to the family of G protein-coupled receptors. (c) The “insulin-like growth factor receptor signaling pathway” is a process crucial to the survival and growth of cells.

When comparing the GO-BP terms of the acoustic system with those of the motor system, the GO terms in the motor system in PoS appear much less specific than those of the auditory system; the significance level of the Benjamini–Hochberg test for the key proteins in the PoS process is lower. This could be an expression of the fact that acoustic signals in the PoS process are perceived as foreign, perceived as non-specific, or not perceived at all by the motor system.

#### 2.2.3. Interactions of Key Proteins with Neurotransmitter Receptors

The TACR3 receptor appears in the gene list of the NH process, but no interactions were observed between this receptor and key proteins (Figure 10A). TACR3 (Tachykinin Receptor 3) belongs to a family of receptors for tachykinins, a neuropeptide involved in neuronal signaling, including roles in pain transmission and the stress response. TACR3 binds with substance P, a specific tachykinin peptide, which plays a role in transmitting pain signals.

Three types of receptors appear in the AS gene list: serotonin, dopamine, and GABA receptors, which show close interactions (Figure 10B). The serotonin receptors provide a very complex regulation of synaptic transmission involving excitation and inhibition [38]. DRD2 (Dopamine Receptor D2) and DRD3 are involved in movement, cognition, and emotional and reward-related dopaminergic signaling [39]. TH (tyrosine hydroxylase) is the rate-limiting enzyme in the biosynthesis of dopamine [40]. Together, TH, DRD2, and DRD3 determine the dynamics of dopaminergic transmission in the brain. GABBR1 (Gamma-Aminobutyric Acid Type B Receptor Subunit 1) is an important component of the GABA(B) receptor, which plays a key role in inhibitory synaptic transmission [41].

The TACR1 and TACR2 receptors are present in the Tin gene list, with close interactions with several key proteins via TAC1 (Figure 10C). TAC1 (Tachykinin Precursor 1) is a precursor protein for several neuropeptides, including Substance P [42]. TAC1 shows on the one hand multiple interactions to key proteins and on the other to TACR1 and TACR3. TACR1 primarily binds Substance P, which mediates processes like pain or stress [43]; TACR3 binds Neurokinin B, another tachykinin, and is involved in several central nervous system functions [44]. The distinct ligand preferences and their signaling pathways may determine their function in different contexts.

## 3. Discussion

### 3.1. Methods Rationale

The identification of candidate key proteins and the analysis of molecular networks has become increasingly possible through the methods of bioinformatics and extensive freely accessible databases. The characterization of the interactions between proteins in the form of protein–protein networks can be helpful in understanding the pathomechanism of tinnitus in the individual brain centers. Proteins (nodes) that have a particularly large number of connections (degrees) to other proteins appear to play a special role in the pathophysiological mechanism of biological processes.

The analyzed genes were extracted from the GeneCards database using appropriate key terms. GeneCards offers the possibility to identify genes that are closely related to biological processes, cellular structures, and diseases; at present, 3170 genes are assigned to the keyword tinnitus. This database has not yet been used for the investigation of tinnitus. Our approach consists of the following steps: (a) Genes with appropriate keywords were selected from the GeneCards database (number of genes 20–100). (b) The interactions of the corresponding proteins were analyzed with the STRING database and the Cytoscape program. (c) Proteins with a combined score > 400 were analyzed with the DAVID database for the enrichment of GO terms cellular components (CCs) and biological processes (BPs). (d) The top three high-degree proteins (HDPs) and the corresponding high-score interaction proteins (HSIPs) were selected as key proteins, whereby the 90th percentile was used as the cut-off value after analyzing the distribution of the combined score (CS) values in order to limit the study. To characterize the role of these key proteins, the GO-BP terms of these proteins were compared with that of the complete gene lists. (e) The comparison of the results with the result of a region of the thalamus that is not primarily specialized for the perception of auditory stimuli is a further indication of the validity of the results. By combining the results from different databases, which are focused on different biological contexts and indicators, reliable statements on biological processes like the synaptic transmission in tinnitus can be obtained.

### 3.2. Role of BDNF

When verifying the overlap of key proteins of the auditory and motor systems in the AS and Tin processes, only BDNF appears as the top one protein in both the auditory and motor systems. BDNF was also identified as the top one key protein for synaptic transmission in the SG, CN, and IC [6,7,8]. BDNF is a universal regulator of brain activity and synaptic transmission [45]; it regulates fundamental processes of survival, differentiation, and growth in the brain. It is an important regulator of synaptic transmission as well as long-term potentiation (LTP) and long-term depression (LTD). The effects of BDNF on LTP and LTD are exerted via TrkB (tropomyosin-related kinase B) receptors [46]. In the thalamus, BDNF has been demonstrated to be an important regulating factor in sensory processing and in the thalamocortical circuit [47,48,49]. This has been shown for the visual system but also for the motor system (dorsolateral geniculate nucleus, a subcortical visual structure in the thalamus). BDNF is involved in excitatory and inhibitory synaptic transmission, in adaptation to sensory experience, and in activity-dependent plasticity [45]. BDNF, with its eight exons and four different promoters, plays an important role in synaptic transmission and plasticity in the brain [50].

### 3.3. Key Proteins and Biological Processes in the Auditory System

#### 3.3.1. Normal Hearing

The top one HDP BDNF shows close interactions with NTRK2 and GDNF. NTRK2 (Neurotrophic Receptor Tyrosine Kinase 2, also known as TrkB) is a receptor for BDNF [48]; it is involved in the regulation of neuronal differentiation, survival, and synaptic plasticity. GDNF (Glial Cell Line-derived Neurotrophic Factor) is a neurotrophic factor that promotes the survival of neurons and the stability of synapses [48]. The top two HDP PVALB (parvalbumin) is a calcium-binding protein that is mainly found in GABAergic interneurons. It plays an important role in maintaining the excitation–inhibition balance in the brain [45]. The top three HDP is SNAP25 (synaptosomal-associated protein 25); it is involved in the release of neurotransmitters from synaptic vesicles [51,52]. Its closest interaction exists with CACNA1A, the calcium voltage-gated channel subunit alpha1A, which mediates the entry of calcium ions into excitable cells.

The GO-BP shows that during normal hearing, two main activities take place in the auditory system of the thalamus (Figure 4A): (a) the negative regulation of neuronal apoptotic process, including the cellular functions that are necessary for cell survival and nervous system development (green), and (b) signal transduction at the level of the synapse or several signaling systems. All processes are focused on the ‘sensory perception of sound’; this pattern of activity corresponds to a normal state of functional connectivity in neuronal activity.

#### 3.3.2. Acoustic Stimulation

The top one HDP BDNF shows close interactions with NTRK2 (see above) and CREB1. CREB1 (cAMP Response Element-Binding Protein 1) is a transcription factor that promotes the transcription of genes important for synaptic plasticity and LTP [53]. The top two HDP GFAP (Glial Fibrillary Acidic Protein) is a protein that is mainly found in astrocytes. It is important for the structural integrity and function of astrocytes, which play a supporting role in synaptic transmission [54,55]. GFAP shows the closest interaction with SYP (synaptophysin); it is a membrane protein of synaptic vesicles and regulates the recycling of vesicles after neurotransmitter release: the mechanism is largely unclear. SYP is also detectable as the top three HDP; it shows the closest interactions with NCAM1 (neural cell adhesion molecule 1). NCAM 1 is a cell-adhesion protein and is involved in the modulation of synaptic transmission [56].

The identified GO-BPs reflect the perception of acoustic signals as xenobiotic signals while ensuring the survival of the cells and integrating them into the circadian rhythm. The key protein list stresses the importance of the BDNF and NTRK2 signaling pathway for neuronal survival, auditory processing, and adaptive changes under conditions of acoustic stimulation. The circadian rhythm seems to be important for the type and efficiency of the response to unexpected acoustic stimulation [57]. Pharmacokinetics provides several examples of the regulation of the effect of drugs that depend on the time of day. Furthermore, the circadian rhythm plays an important role for homeostasis in the CNS [58]. The GO term “positive regulation of miRNA transcription” indicates the role of microRNA in the thalamus (see later).

#### 3.3.3. Tinnitus

In the Tin process, BDNF (top one HDP) shows close interactions with NTRK1 and NTRK3, in contrast to NTRK2, which functions as the most important interaction partner for BDNF in both NH and AS processes. NTRK1 (TrkA) is mainly activated by nerve growth factor (NGF) and NTRK3 (TrkC) by neurotrophin-3 (NTF-3). These differences can also be seen in the SG [6], CN [7], and IC [8]. However, the differences in the molecular effects are not only due to the different ligands but also to the downstream signaling pathways, which have a decisive influence on the plasticity of neurons. NTF3 (neurotrophin-3) is a neurotrophic factor that plays an important role in synaptic transmission by supporting the function and stability of synapses [59,60]. Other close interaction partners of BDNF in the Tin process include GDNF (see above) and CREB1 (see above). The top two HDP is GFAP (Glial Fibrillary Acidic Protein), which is also one of the key proteins in the AS process (see above): GFAP works closely together with S100B, a calcium-binding protein that is mainly produced by astrocytes [61]. Among other functions, it modulates synaptic plasticity. The top three HDP is TNF (tumor necrosis factor); its HSIPs are IL1B (interleukin 1 beta), IL6 (interleukin 6), JUN (Jun proto-oncogene, AP-1 transcription factor subunit), and CD4 (CD4 molecule). The important role of TNF in tinnitus associated with inflammatory processes involving most of the HSIPs has recently been reviewed [62].

The GO-BPs in tinnitus are characterized by two processes: the regulation of transcription by microRNA and the regulation of different developmental processes. The fine-tuning of transcription by microRNAs seems to be essential for the regulation of autophagy and protein homeostasis. Both processes maintain cellular health and stability, protecting against stress and disease development [63]. A role of microRNA was shown in tinnitus [64], in auditory thalamocortical circuit [65], and in auditory cortex in schizophrenia models [39,66]. The developmental processes proliferation and neuron apoptosis indicate a reorganization of synaptic transmission. The evidence of the “cell-surface receptor protein tyrosine kinase signaling pathway”, which includes the proteins BDNF, NTRK3, and NTF3, may transmit signals from extracellular ligands such as growth factors and neurotrophins, intracellular signaling cascades [67], or bottom-up or top-down signals from other auditory centers. Thus, the biological processes in tinnitus reflect processes important for maintaining cellular functions but in an altered state of functional connectivity between neurons.

### 3.4. Key Proteins and Biological Processes in the Motor System

#### 3.4.1. Normal Hearing

In the NH process, it is noticeable that it is not BDNF but AKT1 that acts as the top one HDP. AKT1 is a serine/threonine kinase that plays a central role in the regulation of numerous cellular processes in the brain, in particular LTP [68]. AKT1 shows close interactions with IGF1 (insulin-like growth factor 1), an important protein in the positive regulation of the glycogen biosynthetic process and thus for the maintenance of energy balance [69,70]. IGF1 also plays a crucial role in the regulation of synaptic plasticity and neuronal development. The top two HDP is BDNF, which shows the closest interaction with COMT. BDNF (see above) interacts with COMT (catechol-O-methyltransferase), which is responsible for the degradation of catecholamines, such as dopamine, and thus influences synaptic transmission [71]. The top three HDP is SOD1, which shows close interactions with TARDBP. SOD1 (Superoxide Dismutase 1) is an enzyme that plays an important role in protecting neurons from oxidative stress. TARDBP (TAR DNA-Binding Protein 43) stands for the protein TDP-43, which plays a role in RNA processing and the transport of mRNA. Both proteins play a role in amyotrophic lateral sclerosis (ALS) [72].

The GO-BPs during normal hearing indicate functions of the thalamus that reflect processes at the level of the whole organism (heat), which are important for physiological functions and the homeostasis of the brain. Six GO-BP termini correspond to activities that are general components of synaptic transmission in CNS tissue. Under conditions of normal hearing, synaptic transmission operates in a “global balanced state” [73].

#### 3.4.2. Acoustic Stimulation

BDNF acts as the top one HDP; the HSIP is COMT (see above). The top two HDP is TH (Tyrosine hydroxylase), and its closest interaction partners are SLC6A3 and SNCA [74]. TH is the enzyme that represents the rate-determining step in the synthesis of dopamine. SLC6A3 encodes the dopamine transporter (DAT), a transmembrane protein that regulates the re-uptake of released dopamine from the extracellular space into the presynapse of neurons. SNCA (alpha-synuclein) is a protein involved in the regulation of synaptic–vesicle transport and neurotransmitter release. The top three HDP is the dopamine D2 receptor (DRD2). This receptor modulates neuronal activity in response to dopamine; SLC6A3 (dopamine transporter) removes dopamine from the synaptic cleft, terminating dopamine action. DRD2 also interacts with COMT [75].

The GO-BPs indicate interactions between proteins on three different levels of biological processes, which reflect a finely tuned motor response to acoustic stimulation. (a) Acoustic stimulation is perceived as a “xenobiotic stimulus” (see above), as in the auditory system. (b) Acoustic stimulation triggers the activation of the dopaminergic system at the level of synaptic transmission in the motor system (four GO termini). (c) Processes at the level of the brainstem are set into a state of activity that is an adequate response to an unknown stimulation of the acoustic system. The changes in dopamine signaling in the thalamus were also observed in schizophrenic patients [65].

#### 3.4.3. Tinnitus

The top one HDP is BDNF; it shows close interactions with MAPK3 (mitogen-activated protein kinase 3). MAPK3 is part of a signaling cascade that regulates various processes, such as neuroprotection and neuroplasticity, but also proliferation and differentiation at the cellular level [76]. The interaction between BDNF and MAPK3 can be initiated, e.g., via NTRK2. The top two HDP is AKT1, which shows close interactions with four proteins: MAPK1, MAPK3 (see above), NOS2, and IGF1. MAPK1 (mitogen-activated protein kinase 1, also known as ERK2) is part of the MAP kinase pathway and plays a central role in cell signaling. NOS2 (nitric oxide synthase 2, also known as iNOS (inducible nitric oxide synthase) is an enzyme that produces nitric oxide (NO). NO acts as a neurotransmitter and plays an important role in neuronal communication. Insulin-like growth factor 1 (IGF1) activates the AKT1 signaling pathway and influences cell growth and survival. AKT1, MAPK1, and MAPK3 work together in the PI3K/AKT/MAPK signaling pathway, an important intracellular signaling pathway for many cell processes, including cell growth, proliferation, and survival [76]. The top three HDP is TH (see above), which is detectable as the top two HDP in AS. There is a close interaction between TH and SNCA, which is based on the interaction of dopamine and SNCA [77]. SNCA (alpha-synuclein) is a protein that is found in the presynaptic terminals of neurons and plays a role in the regulation of synaptic-vesicle transport and the release of neurotransmitters.

The GO-BP termini reflect interactions of proteins that are involved in three different categories: (a) Tinnitus is perceived in the MoS as “xenobiotic”, with changes corresponding to the GO termini “response to cadmium” or cocaine [78]. (b) The chemical synaptic transmission is not, however, associated with the activation of the dopamine system but with phosphorylation (as in normal processes; see above); the “tachykinin receptor signaling pathway” occurs in parallel. Tachykinins (including substance P) bind to special receptors and use them to transmit signals. These processes play a role, for example, in the mediation of pain and in inflammatory reactions. Tachykinins belong to neuropeptides; they bind to tachykinin receptors and activate G proteins [42,79]. The GO process “insulin-like growth factor receptor signaling pathway” indicates processes of cell growth, survival, and metabolism. The GO terms “response to cocaine, fungi, cadmium” are uncharacteristic and may be non-specific terms among the 209 significant chart records (Figure 9C).

### 3.5. Synaptic Transmission in the Auditory and Motor Systems

#### 3.5.1. Auditory System

The interaction of key proteins and neurotransmitter receptors appears to have a significant influence on synaptic transmission under the respective conditions (NH, AS, Tin).

In the NH process, GRIA3 shows close interactions with key proteins; it is important for rapid excitatory, timely signaling to the cortex, a property that is particularly important for language processing [80]. The nicotinic acetylcholine receptors CHRNA9 and CHRNA10ACH are present in the PPI network but show no interactions with key proteins. Both are also involved in auditory processing, but they are not influenced by key proteins under the conditions of normal hearing and show normal physiological activity.

Under the conditions of AS, the GRIN1 subunit of the NMDA receptor shows numerous interactions with key proteins; the CHRNA4 protein (alpha4 subunit of nicotinic acetylcholine receptors; nAChRs) shows interactions with GRIN1. Similar to GRIA3, GRIN1 is also involved in excitatory auditory processing, but more in the sense of modulating complex acoustic stimuli [81]. CHRNA4 primarily influences processes of attention and sensory processing [82,83,84]. These interactions are important for the organism’s response and adaptation to AS conditions that require a high level of attention and response.

Under Tin conditions, two proteins show close interactions with key proteins: GRIN1 and PDYN. The appearance of GRIN1 shows that bottom-up signaling acts in the auditory system of the thalamus, similar to AS, but the processing is completely different. GRIN1 shows close interactions with PDYN, which acts on kappa-opioid receptors and modulates neural excitability [85]. This leads to an imbalance between excitation and inhibition. Dynorphins influence stress responses and sensory processing and can contribute to tinnitus perception in the cortex [85,86,87].

#### 3.5.2. Motor System

In contrast to the AuS, different proteins are under the influence of the respective key proteins in the MoS in the NH, AS, and Tin processes. In NH, the receptor TACR3 (Tachykinin Receptor 3) is detectable in the PPI network without interactions with key proteins. TACR3 may be involved in excitatory signaling, affecting different neurons with specific functions [88].

In the AS process, all detectable receptors (serotonin, dopamine, and GABA receptors) show a multitude of interactions with key proteins. These interactions enable finely tuned signaling for motor reactions in the sense of preventing uncontrolled hyperactivity and organizing defenses or escape.

In Tin, it is primarily TACR1 (receptors for tachykinins, such as substance P) and TACR3 (receptor for neurokinin B) that show close and numerous interactions with the corresponding key proteins via TAC1. Under normal conditions, substance P was classified as a regulatory peptide because it is capable of normalizing responsiveness to pain [89,90]. The interactions of TACR1 and TACR3 in the motor system of the thalamus appear to have an inhibitory effect under the conditions of tinnitus, thereby attenuating, inhibiting, or normalizing the influence of tinnitus signals that are perceived in the AuS. Several findings from other systems indicate an inhibitory effect of tachykins, and the effect appears to be context dependent [43,91,92,93].

### 3.6. Limitations

The limits of the approach we use have been described in detail in previous work [6,7,8]. The procedure is based on the evaluation of databases, which of course can only cover current knowledge. The findings and hypotheses developed are proposals for experimental projects. Both the search for suitable keywords and the assignment of GO processes to gene lists and the analysis of interactions are associated with errors. It should be emphasized that this study involves the evaluation of results from biostatistical databases only; the results of our study must be experimentally validated. We are currently unable to carry out these investigations. However, the results of the study are suitable as a guide for targeted experimental investigations. With the use of appropriate key words, it is possible to identify key proteins for biological processes from the large, publicly accessible databases [94]. The methods used in this study provide information on key proteins that is consistent with the current state of knowledge on tinnitus and the function of the thalamus.

## 4. Material and Methods

The analysis of key proteins and biological processes according to gene ontology (GO-BPs) in auditory centers of the brain may provide insights into synaptic transmission at the molecular level [20]. To better understand the processes of synaptic transmission in the thalamus, in the present study, we compared key proteins and the occurrence of GO-BPs under the conditions of normal hearing (NH) and acoustic stimulation (AS) with those occurring in tinnitus (Tin) patients. Three gene lists were compiled from the GeneCard database [21] (https://www.genecards.org/; accessed on 13 March 2025) for the following keywords: (a) “synaptic transmission”, thalamus, “auditory system” or “medial geniculate nucleus”, “normal hearing” or “perception of sound” (NH; there are not enough data available for the term “medial geniculate nucleus”); (b) “synaptic transmission”, thalamus, “auditory system” or “medial geniculate nucleus”, “acoustic stimulation” or “acoustic stimuli” (AS); and (c) “synaptic transmission”, thalamus, “auditory system” or “medial geniculate nucleus”, tinnitus (Tin). Briefly, these gene lists were characterized by analyses of gene overlap using Venn diagrams (http://bioinformatics.psb.ugent.be/webtools/Venn/, accessed on 13 March 2025). The construction of protein–protein interaction networks (PPI) was performed using the STRING database [22] (Search Tool for the Retrieval of Interacting Genes; https://string-db.org/). For protein-enrichment analysis, we used the Database for Annotation, Visualization, and Integrated Discovery (DAVID) [23] (https://david.ncifcrf.gov/); Fisher’s exact test and the Benjamini–Hochberg value were used for the significance of the GO terms (*p* < 0.01 for the gene lists and *p* < 0.05 for the key word lists). Any deviations in the value for the multiple testing corrections for key proteins are labelled separately in the legend. GO-BP terms with a Benjamini–Hochberg value *p* > 0.05 were included in the corresponding figures to illustrate the pathophysiological similarity of the annotations by the gene lists and the key protein lists. The Cytoscape data analyzer (https://cytoscape.org/) was used to identify key proteins in the PPI network. For identification of key proteins, the following criteria were used: (a) for nodes, the number of degrees, the clustering coefficient, the closeness centrality, and (b) for edges, the coexpression, the experimentally determined interaction, automated text mining, and the combined score (CS). Because of different biases within the list of proteins and to limit the study, only the top three high-degree proteins (HDPs) were selected for analysis. We hypothesized that the HDPs and the corresponding HSIPs, together named key proteins, play a functionally important role in the regulation of protein–protein networks. To test the specificity of our approach, we compared the results obtained for the auditory system of the thalamus (AuS) with those obtained for the motor system (MoS) of the thalamus. We analyzed comparable gene lists, with the one difference that we replaced the term ‘auditory system’ with the term ‘motor system’. The following databases served for the brief definition and characterization of proteins or genes: https://www.ncbi.nlm.nih.gov/; https://www.uniprot.org/uniprotkb/; https://syngoportal.org; https://thebiogrid.org/, accessed on 13 March 2025.

## 5. Conclusions

Under normal hearing conditions, important GO-BPs in AuS activities are focused on the ‘sensory perception of sound’, and in the MoS processes, they are associated with functions at the level of the whole organism. This pattern of activity corresponds to a normal state of functional connectivity. Under conditions of acoustic stimulation in the AuS and MoS, GO-BPs suggest the perception of stimulated acoustic signals as a “xenobiotic” signal while ensuring the survival of the cells. In the AuS, GO-BPs are classified into a response according to the circadian rhythm, and in the MoS, in the regulation of the dopaminergic system. GO-BPs under conditions of tinnitus indicate in the AuS two processes: the regulation of transcription by microRNA and the regulation of different developmental processes. The fine-tuning of transcription by microRNAs seems to be essential for the regulation of autophagy and protein homeostasis. Tinnitus also is perceived in the MoS as “xenobiotic”, a possible inhibitory effect of tachykinins that may attenuate or inhibit the influence of tinnitus. The identification and evaluation of key proteins is useful for confirming, supplementing, and specifying the GO termini obtained from complete gene lists. When sufficiently reliable data on the selected wording and the analyzed disease and comparative states are available, the method could be extended to other diseases and brain regions.

## Figures and Tables

**Figure 1 ijms-26-05804-f001:**
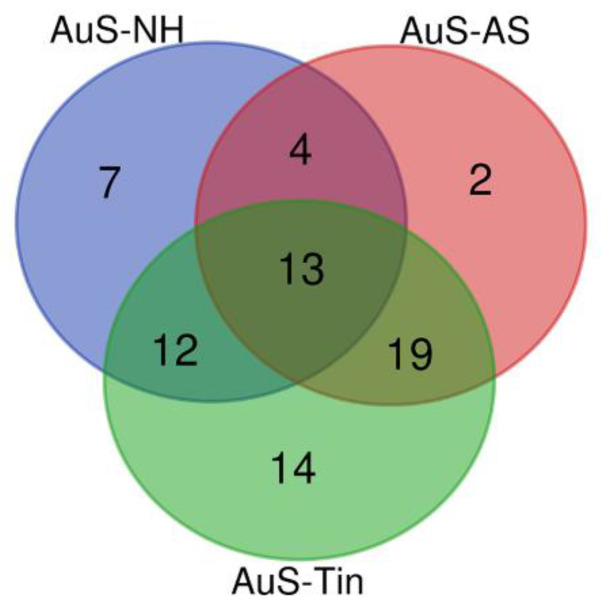
The Venn diagram of the genes involved in normal hearing (NH, *n* = 36), acoustic stimulation (AS, *n* = 38), and tinnitus (Tin, *n* = 58). The overall number of unique elements *n* = 71.

**Figure 2 ijms-26-05804-f002:**
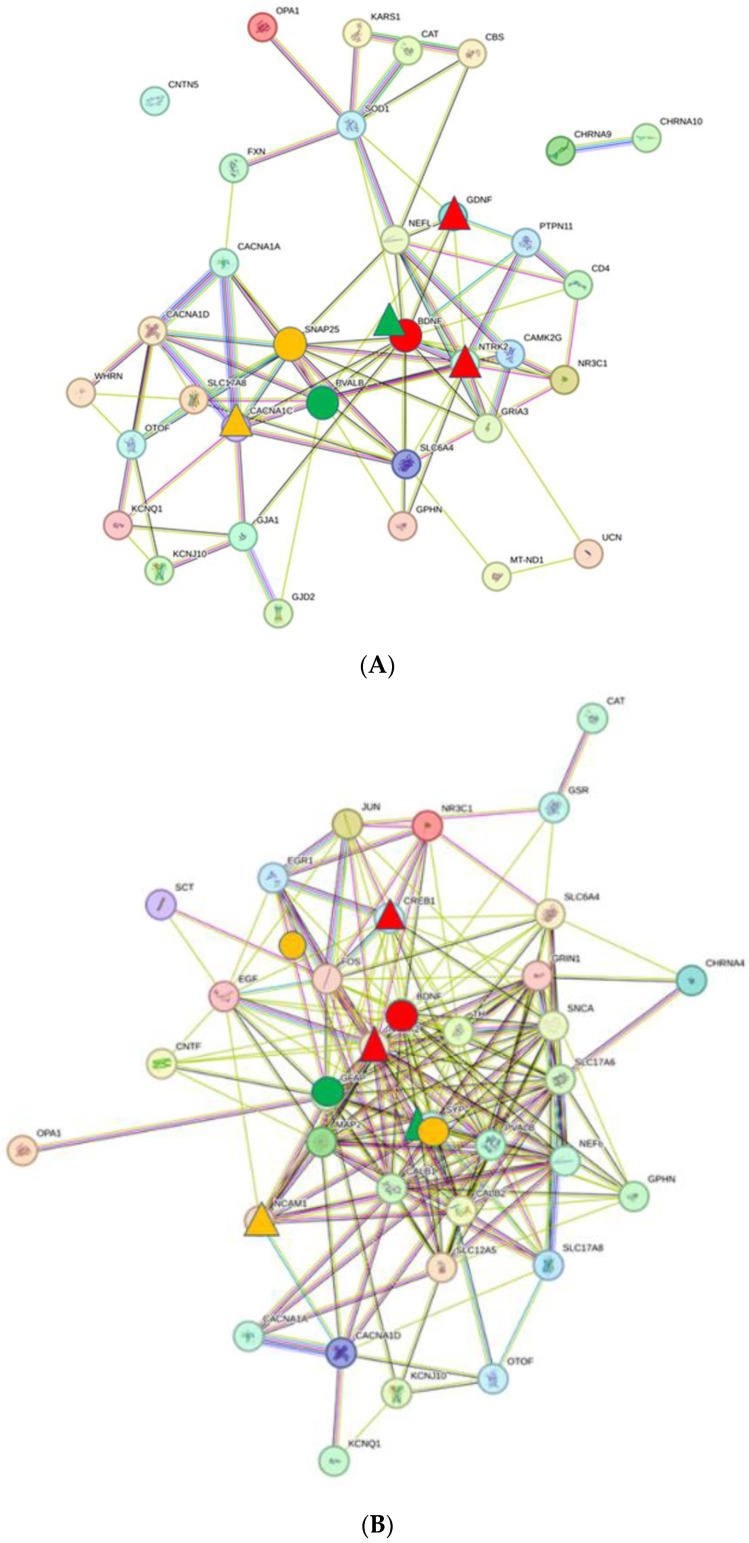
PPI networks of the normal hearing (NH, (**A**)), the acoustic stimulation (AS, (**B**)), and the tinnitus (Tin, (**C**)) processes in the auditory system of the thalamus. Topological criteria—see text. The key proteins in the networks are labeled: Red circle—top 1 high-degree protein (HDP); red triangles—high-score interaction proteins (HSIPs). Green circle—top two HDPs; green triangles—corresponding HSIPs. Yellow circle—top 3 HDP; yellow triangles—corresponding HSIPs.

**Figure 3 ijms-26-05804-f003:**
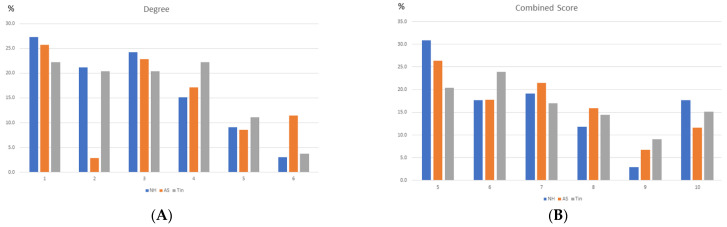
The frequency distribution of the degree (**A**) and CS values (**B**) of the NH, AS, and Tin networks. The frequency distribution of the degree and CS values were calculated as percentages of the number of nodes and edges. Group 6 of the degree distribution corresponds to the top 16.7% of the degree values. The CS values cover a range of 400–999. Group 6 of the CS distribution corresponds to CS values between 901 and 999, group 5 corresponds to CS values of 801–900, etc.

**Figure 4 ijms-26-05804-f004:**
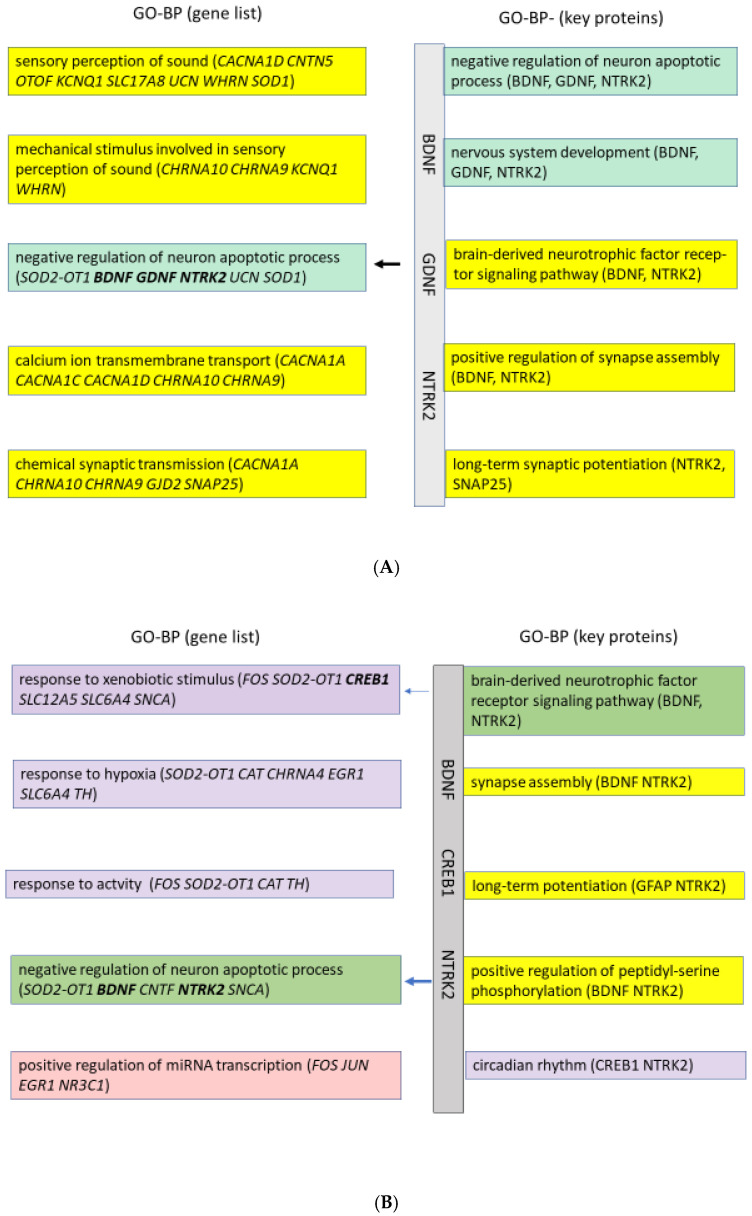
(**A**): Biological processes (GO) of the *normal hearing* process in the auditory system of the thalamus. The order is according to statistical significance. Colors were selected according to general functions: yellow—signal systems; green—developmental processes (including growth, apoptosis). Characteristic values of the GO terms: The gene list (36 IDs, 125 chart records, *p* < 0.01): the percentage of the proteins involved in the GO term 11–22; *p*-values 1.2E-8 to 6.8E-4; fold enrichment 12–117. The key protein list (6 IDs, 22 chart records, *p* < 0.05): the percentage of the proteins involved in the GO term 33–50; *p*-values 6.2E-4 to 1.7E-2; fold enrichment 22–120. The Benjamini–Hochberg value for GO-BP terms (key proteins) = 4.9E-1 to 8.3E-2. (**B**): Biological processes (GO) of the *acoustic stimulation* process in the auditory system of the thalamus. The order is according to statistical significance. Colors: violet—response to xenobiotic signals; green—developmental process (apoptosis); red—miRNA transcription; yellow—signal systems. The gene list (38 IDs, 95 chart records, *p* < 0.01): the percentage of the proteins involved in the GO term 11–18; *p*-values 6.0E-6 to 1.9E-4; fold enrichment 15–41. The key protein list (6 IDs, 17 chart records, *p* < 0.05): the percentage of the proteins involved in the GO term 33.3; *p*-values 1.4E-2 to 1.8E-2; fold enrichment 92–104. The Benjamini–Hochberg value for GO-BP terms (key proteins) = 6.0E-1. (**C**): Biological processes (GO) of the *tinnitus* process in the auditory system of the thalamus. Red—miRNA transcription; yellow—signal systems; green—developmental processes (negative regulation of apoptosis); violet—response to xenobiotic stimulus. The gene list (58 IDs, 236 chart records, *p* < 0.01): the percentage of the proteins involved in the GO term 12–16; *p*-values 1.7E-8 to 5.7E-6; fold enrichment 13–41. The key protein list (10 IDs, 111 chart records, *p* < 0.05): the percentage of the proteins involved in the GO term 25–50; *p*-values 4.8E-7 to 1.7E-5; fold enrichment 22–347. The Benjamini–Hochberg value for GO-BP terms (key proteins) = 2.3E-4 to 4.9E-4. The arrows indicate GO terms that include key proteins.

**Figure 5 ijms-26-05804-f005:**
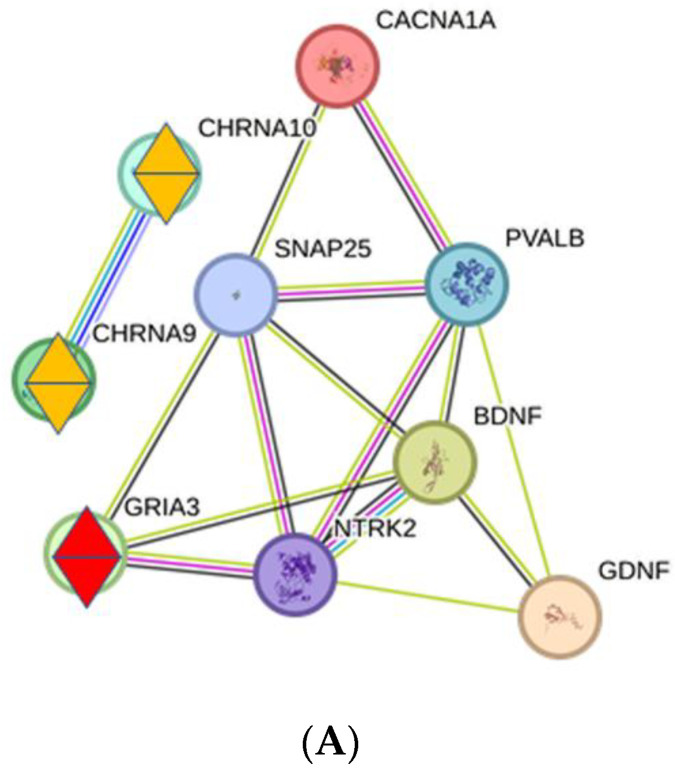
(**A**): Interactions between key proteins and neurotransmitter receptors under normal hearing conditions. Neurotransmitter receptors are marked with two triangles; red—excitatory, yellow—modulatory. (**B**): Interactions between key proteins and neurotransmitter receptors under conditions of acoustic stimulation. Neurotransmitter receptors are marked with two triangles; red—excitatory, yellow—modulatory. (**C**): Interactions between key proteins and neurotransmitter receptors in the AuS in tinnitus. Neurotransmitter receptor GRIN1 is marked with two red triangles; PDYN (purple polygon) interacts closely with GRIN1 (CS = 930), with BDNF (CS = 582) and CREB1 (CS = 571).

**Figure 6 ijms-26-05804-f006:**
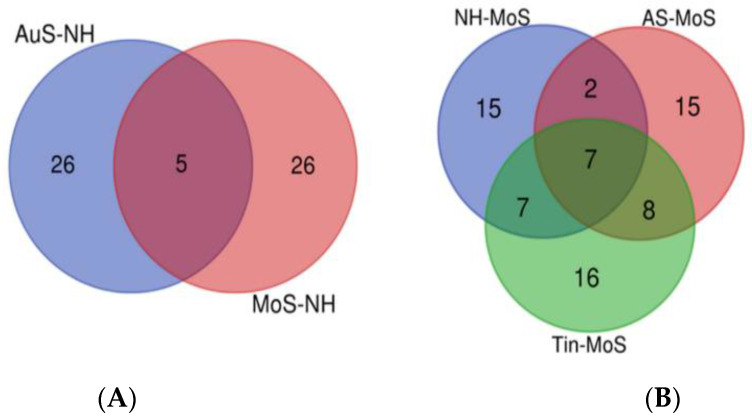
(**A**): The Venn diagram of the genes present in the auditory system (AuS) and the motor system (MoS) of the thalamus under conditions of normal hearing. The overall number of unique elements: AuS—65, MoS—70. (**B**): Genes involved in normal hearing (NH, *n* = 31), acoustic stimulation (AS, *n* = 32), and tinnitus (Tin, *n* = 38) in the MoS of the thalamus.

**Figure 7 ijms-26-05804-f007:**
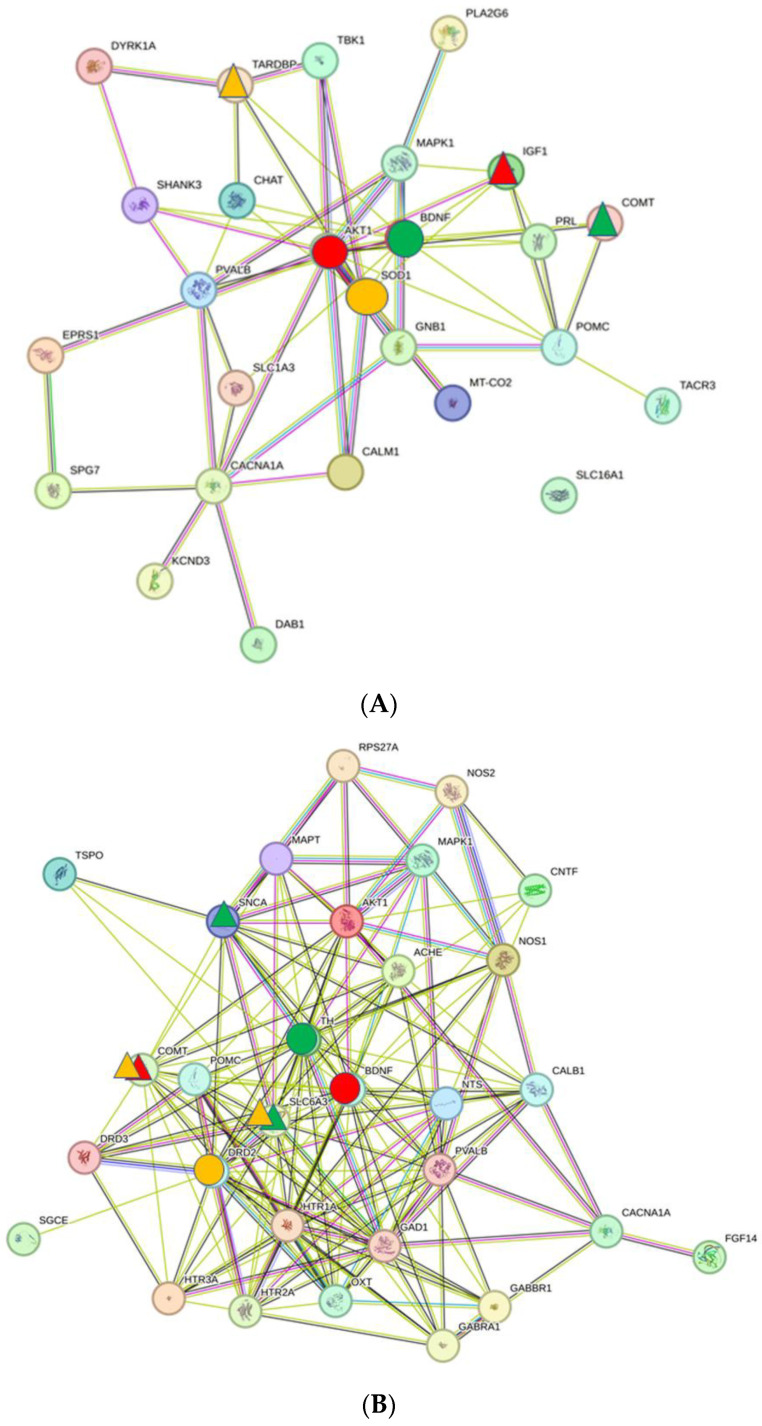
The PPI network of the NH (**A**), AS (**B**), and Tin (**C**) processes in the motor system of the thalamus. For the topological criteria, see the text. The key proteins are labeled in the networks. The key proteins in the networks are labeled: Red circle—top 1 high-degree protein (HDP); red triangles—high-score interaction proteins (HSIPs). Green circle—top 2 HDP; green triangles—corresponding HSIPs. Yellow circle—top 3 HDP; yellow triangles—corresponding HSIPs.

**Figure 8 ijms-26-05804-f008:**
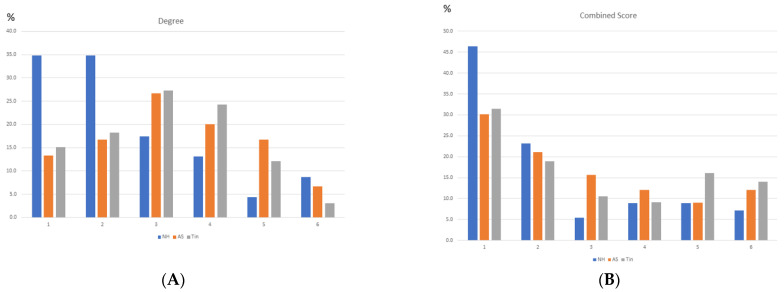
The frequency distribution of the degree (**A**) and CS (**B**) values of the NH, AS, and Tin networks. The frequency distribution of the degree and CS values were calculated as percentages of the number of nodes and edges. Group 6 of the degree distribution corresponds to the top 16.7% of the degree values. Group 6 of the CS distribution corresponds to CS values of 901–999, group 5 correspond to CS values of 801–900, etc.

**Figure 9 ijms-26-05804-f009:**
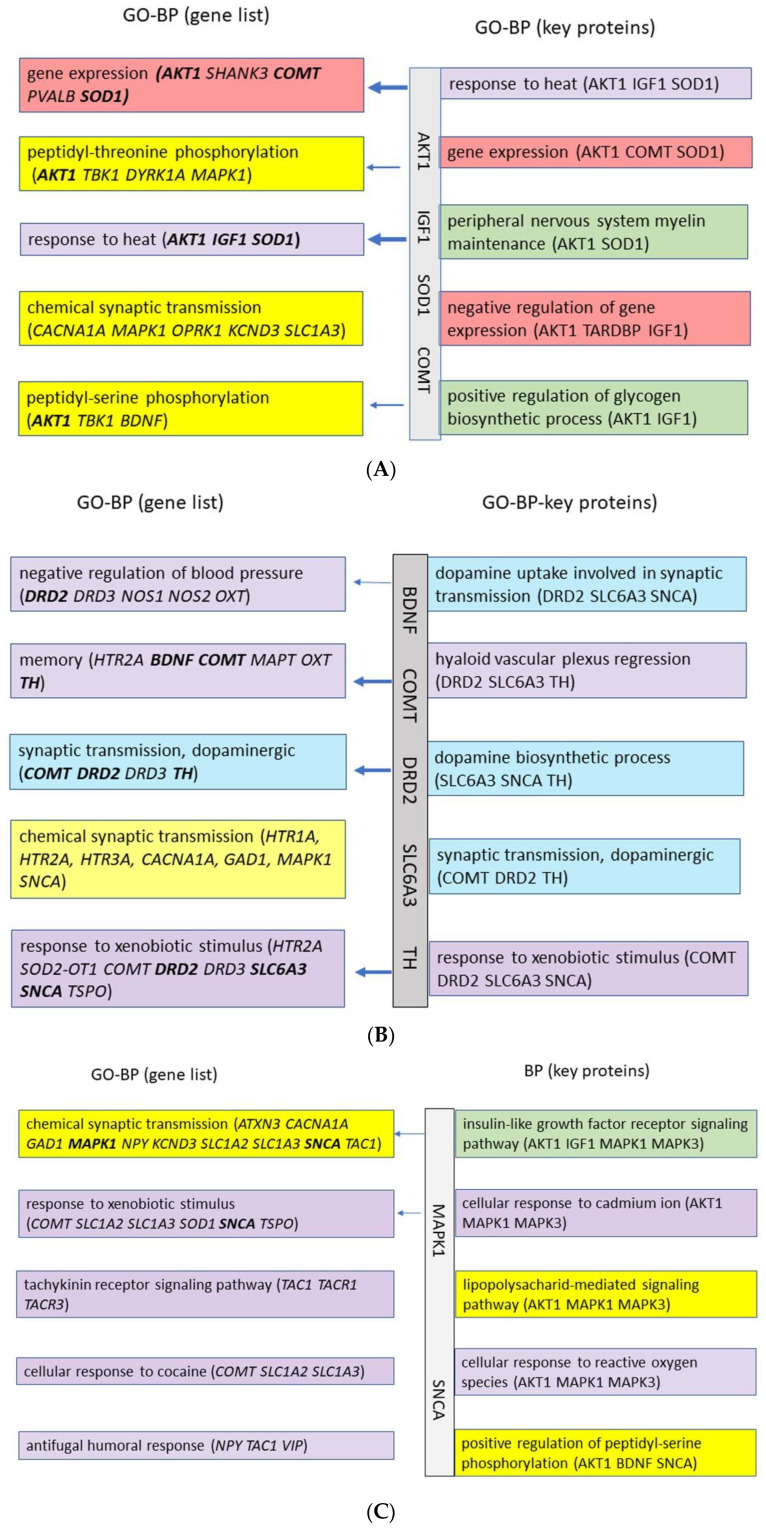
(**A**): The biological processes (GO) of the normal hearing process in the motor system of the thalamus. The order is according to statistical significance. Colors selected according to general functions. Red—gene expression; yellow—signal systems; violet—uncharacteristic GO terms; green—biosynthetic processes. The gene list (25 IDs, 44 chart records, *p* < 0.01): the percentage of the proteins involved in the GO term 12–20; *p*-values 5.3E-5 to 3.4E-3; fold enrichment 13–50. The key protein list (6 IDs, 33 chart records, *p* < 0.05): the percentage of the proteins involved in the GO term 33–50; *p*-values 4.1E-5 to 4.4E-3; fold enrichment 31–719. The Benjamini–Hochberg value for GO-BP terms (key proteins) = 1.3E-2 to 2.7E-1. (**B**): The biological processes (GO) of the acoustic stimulation process in the motor system of the thalamus. Violet—response to signals from the auditory system; blue—response of the dopaminergic system; yellow—signal system. The gene list (30 IDs 113 chart records, *p* < 0.01): the percentage of the proteins involved in the GO term 13–23; *p*-values 9.3E-8 to 1.9E-6; fold enrichment 18–216. The key protein list (6 IDs, 57 chart records, *p* < 0.05): the percentage of the proteins involved in the GO term 50–67; *p*-values 8.0E-7 to 2.2E-5; fold enrichment 50–1618. The Benjamini–Hochberg value for GO-BP terms (key proteins) = 1.2E-3 to 1.6E-4. (**C**): The biological processes (GO) of the tinnitus process in the motor system of the thalamus. Yellow—signal systems; violet—response to non-specific signals; green—growth factor signaling. The gene list (33 IDs, 121 chart records, *p* < 0.01). The percentage of the proteins involved in the GO term 9–30; *p*-values 9.0E-11 to 1.7E-4; fold enrichment 14–221. The key protein list (8 IDs, 209 chart records, *p* < 0.05): the percentage of the proteins involved in the GO term 38–50; *p*-values 2.4E-6 to 2.7E-4; fold enrichment 104–208. The Benjamini–Hochberg value for GO-BP terms (key proteins) = 9.9E-4 to 1.5E-1. The arrows indicate GO terms that include key proteins.

**Figure 10 ijms-26-05804-f010:**
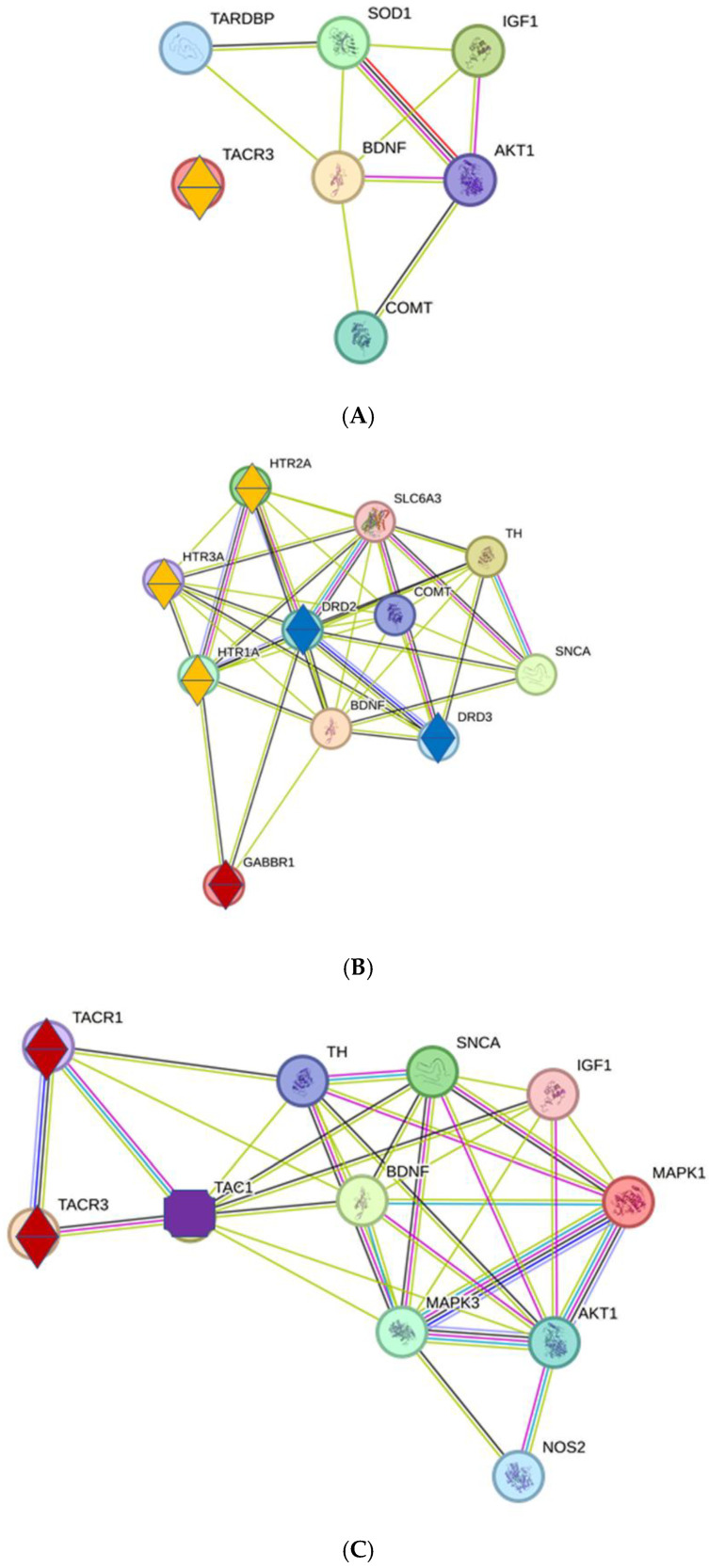
(**A**): Interactions between key proteins and neurotransmitter receptors under normal hearing conditions. The receptor for TAC is marked with two triangles; yellow—modulatory. (**B**): Interactions between key proteins and neurotransmitter receptors under conditions of acoustic stimulation. Neurotransmitter receptors are marked with two triangles; yellow—modulatory; blue—dopaminergic; brown—GABA-ergic. (**C**): Interactions between key proteins and neurotransmitter receptors in tinnitus. Neurotransmitter receptors are marked with two triangles; brown—modulatory or inhibitory. TAC1 (purple polygon) mediates the effects of key proteins with TAC receptors.

**Table 1 ijms-26-05804-t001:** Key proteins in the PPI network of the auditory system of the thalamus in the normal hearing (NH), acoustic stimulation (AS), and tinnitus (Tin) groups.

HDP	Clus	Clos	Degree	HSIP	Coex	Exp	Text	CS	EB
**NH**									
BDNF	0.32	0.65	15	NTRK2	60	691	999	999	14.3
				GDNF	48	0	963	963	14.9
PVALB	0.33	0.57	12	BDNF *	47	0	774	776	18.5
SNAP25	0.44	0.58	11	CACNA1A *	166	0	857	875	24.1
**AS**									
BDNF	0.53	0.76	24	NTRK2	60	961	999	999	2.7
				CREB1	0	0	956	956	4.8
GFAP	0.53	0.72	22	SYP *	239	0	826	860	6.2
SYP	0.62	0.68	20	NCAM1 *	184	54	880	899	6.9
**Tin**									
BDNF	0.45	0.76	37	NTRK3	65	65	999	999	21.7
				NTRK1	60	65	999	999	10.1
				NTF3	82	958	774	991	12.0
				GDNF	48	0	963	963	6.5
				CREB1	0	0	956	956	5.6
GFAP	0.49	0.72	33	S100B *	238	87	901	925	5.6
TNF **	0.53	0.67	28	IL1B	616	0	993	998	4.5
				IL6	261	0	989	994	4.3
				JUN	151	0	881	989	3.3
				CD4	92	0	966	968	7.0

HDP—High-degree protein, Clus—clustering coefficient, Clos—closeness centrality, HSIP—high-score interaction protein, Coex—coexpression, Exp—experimentally determined interaction, Text—automated text mining, CS—combined score, EB—edge betweenness. * Below the critical values of CS > 90th percentile for NH > 937; AS > 904; Tin > 935. ** TH and SYP also show degree values of 28 but with only one HSIP (TH-SNCA and SYP-SNAP25).

**Table 2 ijms-26-05804-t002:** Top five GO terms for cellular components (CCs) for NH, AS, and Tin processes in the auditory system of the thalamus on the basis of the gene lists (Appendix A, Table A1, Table A2 and Table A3) and the key protein lists (Table 1).

Normal Hearing (NH) *Gene List*: 36 IDs, 125 Chart Records, *p* < 0.01	Acoustic Stimulation (AS) *Gene List*: 38 IDs, 95 Chart Records, *p* < 0.01	Tinnitus (Tin) *Gene List:* 58 IDs, 236 Chart Records, *p* < 0.01
*Significance: 2.3E-6 to 3.1E-4* -synapse (9 *) -monoatomic ion channel complex (5) -dendrite (6) -neuron projection (6) -axon (6)	*Significance: 1.7E-9 to 2.3E-6* -neuron projection (10) -terminal bouton (6) -dendrite (10) -synapse (10) -axon (8)	*Significance: 4.3E-13 to 1.2E-7* -neuronal cell body (15) -dendrite (14) -neuron projection (11) -monoatomic ion channel complex (7) -terminal bouton (6)
*Key protein list*: 6 IDs, 22 chart records, *p* < 0.05	*Key protein list:6 IDs, 17 chart records, p <* 0.05	*Key protein list:* 13 IDs, 118 chart records, *p* < 0.05
*Significance: 3.0E-3 to 3.1E-2* -axon (3) -synapse (3) -perinuclear region of cytoplasm (3) -cytoplasm (5) -synaptic vesicle (2)	*Significance: 1.2E-2 to 3.1E-2* -terminal bouton (2) -perinuclear region of cytoplasm (3) -synaptic vesicle (2)	*Significance: 4.3E-4 to 1.1E-3* -extracellular space (7) -axon (4)

The GO terms are ordered according to the *p*-values. * The number of proteins involved in the GO term.

**Table 3 ijms-26-05804-t003:** Key proteins in the PPI network of the motor system of the thalamus in the NH, AS, and Tin groups.

HDP	Clus	Clos	Degree	HSIP	Coex	Exp	Text	CS	EB
**NH**									
AKT1	0.22	0.71	14	IGF1	0	63	934	935	17.8
BDNF	0.30	0.62	12	COMT	0	0	919	919	16.7
SOD1	0.36	0.53	8	TARDBP	77	0	989	989	15.5
**AS**							919		
BDNF	0.46	0.83	24	COMT	0	0	944	919	5.58
TH	0.54	0.74	20	SLC6A3	147	0	975	950	4.64
				SNCA	0	95	924	997	7.17
DRD2	0.56	0.73	19	COMT	0	0	995	924	6.83
				SLC6A3	54	380		999	5.74
**Tin**							779		
BDNF	0.41	0.71	21	MAPK3	0	0	872	942	10.4
AKT1	0.44	0.68	17	MAPK1	118	110	432	988	10.3
				MAPK3	266	11	775	957	5.4
				NOS2	0	45	934	976	22.8
				IGF1	0	63	975	935	7.08
TH	0.53	0.62	16	SNCA	0	95		997	13.0

HDP—High-degree proteins, Clu—clustering coefficient, Clo–closeness centrality, Degree, HSIP—high-score interaction protein, Coex—coexpression, Exp—experimentally determined interaction, Text—automated text mining, CS—combined score, EB—edge betweenness. Critical values for HSIPs: CS > 90th percentile: NH > 940; AS > 909; Tin > 954.

## Data Availability

Data used can be retrieved from the databases mentioned.

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
