# Peer review of "Candidate Key Proteins of Tinnitus in the Auditory and Motor Systems of the Thalamus"

_ijms, 2025, doi:10.3390/ijms26125804_

Round 1

Reviewer 1 Report

Comments and Suggestions for Authors

The authors explored key proteins and gene ontology biological processes in tinnitus in the thalamic auditory and motor systems. This study provided a bioinformatics-extensive analysis using integrative approaches to delineate PPI networks and extract topologically significant proteins. The study to be more significant and data-rich some interpretive results require strengthening.

Major Concerns:

  1. Validations of Bioinformatics Findings: The authors should include limitations related to lack of experimental validation more critically or provide preliminary experimental data validating a subset of these predicted key proteins in relevant thalamic regions.
  2. Missing rationale behind comparing auditory and motor systems: The authors compare and explain the differences between auditory and motor system however the rationale behind these comparisons is missing. The manuscript should expand on the hypothesis that tinnitus-related alterations in motor thalamus hold functional significance rather than limiting the discussions to observed differences in the PPI network.

Minor Concerns:

  1. Cross validation with independent ontologies like enrichment platforms (gProfiler/Enrichr) or pathway databases (KEGG/Reactome) is recommended to enhance confidence and reduce annotation bias.
  2. The confidence threshold used for PPI construction is missing which affects downstream interpretation.
  3. Network topology metrics like clustering coefficient and closeness centrality should be related to known synaptic properties in tinnitus pathophysiology to add proper biological context to the data.

Author Response

x

Response to Reviewer I (21. May)

Thank you very much for the valuable advice given to improve our manuscript.

Major Concerns:

  1. Validations of Bioinformatics Findings: The authors should include limitations related to lack of experimental validation more critically or provide preliminary experimental data validating a subset of these predicted key proteins in relevant thalamic regions.
  2. Missing rationale behind comparing auditory and motor systems: The authors compare and explain the differences between auditory and motor system however the rationale behind these comparisons is missing. The manuscript should expand on the hypothesis that tinnitus-related alterations in motor thalamus hold functional significance rather than limiting the discussions to observed differences in the PPI network.

Response

  1. The ‘limitations’ section has been revised and the following passage has been added:

"It should be emphasized that this study involves the evaluation of results from biostatistical databases only; the results of our study must be experimentally validated. We are currently unable to carry out these investigations. However, the results of the study are suitable as a guide for targeted experimental investigations”

  1. The rationale behind comparing auditory and motor system are:

  1. Tinnitus is a disease of the auditory system. It is expected that different key proteins occur in the motor system compared to the auditory system; 2. It is known that the dopaminergic system is activated in the thalamus during motor activation. A change in the dopaminergic system only during acoustic stimulation would be an indication of the reliability of the chosen procedure. The inclusion of further systems (sensory, visual) to confirm the reliability and specificity of our approach is beyond the scope of the present work.

As these problems also play a role in review II, we have included the following passage in the discussion to explain our approach:

3.1 Methods applied

            The identification of candidate key proteins and the analysis of molecular networks has become increasingly possible through the methods of bioinformatics and extensive freely accessible databases. The characterization of the interactions between proteins in the form of protein-protein networks can be helpful in understanding the pathomechanism of tinnitus in the individual brain centers. Proteins (nodes) that have a particularly large number of connections (degrees) to other proteins appear to play a special role in the pathophysiological mechanism of biological processes.

            The analyzed genes were extracted from the GeneCards database using appropriate key terms. GeneCards offers the possibility to identify genes that are closely related to biological processes, cellular structures and diseases, at present 3,170 genes are assigned to the keyword tinnitus. This database has not yet been used for the investigation of tinnitus. Our approach consists of the following steps: a.) Selection of genes with appropriate keywords from the GeneCards database (number of genes 20-100). b.) Analyses of the interactions of the corresponding proteins with the STRING database and the Cytoscape program. c.) Proteins with a combined score > 400 were analyzed with the DAVID database for the enrichment of GO-terms cellular components (CC) and biological processes (BP). d.) The top three high degree proteins (HDP) and the corresponding high score interaction proteins (HSIP) were selected as key proteins, whereby the 90th percentile was used as the cut-off value after analyzing the distribution of the combined score (CS) values in order to limit the study. To characterize the role of these key proteins, the GO-BP terms of these proteins were compared with that of the complete gene lists. e.) Comparison of the results with the result of a region of the thalamus that is not primarily specialized for the perception of auditory stimuli is a further indication of the validity of the results. By combining the results from different databases, which are focused on different biological contexts and indicators, reliable statements on biological processes like the synaptic transmission in tinnitus can be obtained.

Minor Concerns:

  1. Cross validation with independent ontologies like enrichment platforms (gProfiler/Enrichr) or pathway databases (KEGG/Reactome) is recommended to enhance confidence and reduce annotation bias.
  2. The confidence threshold used for PPI construction is missing which affects downstream interpretation. Network topology metrics like clustering coefficient and closeness centrality should be related to known synaptic properties in tinnitus pathophysiology to add proper biological context to the data.

Response

  1. We started cross validation by comparing the results of DAVID with Metascape analyses. When comparing DAVID and Metascape data, we observed that the statements of Metascape are more general than those of DAVID. A true comparison of the informative value of both databases requires further and more extensive analyses. It is known that Metascape emphasizes clustering enriched ontology terms, and that DAVID has a higher sensitivity, especially at a low number of proteins (key proteins).

  1. A combined score of >400 was used for the PPI figures.

The relationship between clustering coefficient and closeness centrality and pathophysiological criteria of tinnitus is a very interesting question. No firm conclusions can be drawn from the present study; the answer to this question requires further intensive investigations.

Johann Gross

06.06.2025

Reviewer 2 Report

Comments and Suggestions for Authors

This submission is a comprehensive bioinformatics analysis to identify relevant proteins in tinnitus and examining their interactions, GO terms, and receptor interactions in the auditory and motor systems of the thalamus in 3 conditions: NH, AS, and Tin. The use of multiple databases and appropriately described methods provide rigor to the study. 

This field is currently understudied, and the inclusion of a dual-system (auditory and motor) adds novelty to this article. The tables and figures are also informative and clearly presented. However, there are several sections and issues that need to be addressed.

Major Points

  1. The Abstract should at least cover the background, particularly the rationale/significance of the study before jumping into the methods and results. It's also crucial that the problem statement be indicated first before any discussion of methods and results.
  2. The databases (GeneCard, STRING, DAVID, Cytoscape), the specific search criteria, dataset sizes, and normalization were not discussed in detailed. How were the genes selected for each condition? Please give more context in the selection of thresholds (e.g., CS > 90th percentile). What were the filtering criteria?
  3. Statistical significance was indicated but there was no discussion of multiple testing corrections. Were the p-values corrected for multiple comparisons, particularly in GO term enrichment?
  4. The interpretations of the results seem overreaching, beyond what the data can describe. The descriptions of "xenobiotic response" and "Circadian rhythm" must be taken speculatively as these are merely associated with statistical results. I suggest using the terms "may indicate," "suggests," in connecting these interpretations with the statistical results.
  5. The Discussion and Limitations can be expanded with a discussion on the need for follow-up experimental validation as this article is purely an in-silico approach. 

Minor Points

  1. Due to the variety of the nodes in each figure, I suggest including a figure legend to indicate what each node means for easier tracking of the readers, instead of simply describing what the node represents in the captions. 
  2.  A glossary should be provided to also indicate the several abbreviations used in this article, and to allow readers to track the terms easily.
  3. The last paragraph of the introduction can be transferred to the Discussion or Conclusion since it covered the benefits of an in silico approach to establishing the connections between genes and proteins in these conditions. 
  4. The manuscript's grammar and sentence structuring must be improved further. I suggest keeping sentences simple and short. Due to the complexity of the study, it may be difficult to keep track of the discussion when the sentences are too elaborate. I have some suggestions below for the improvement of Quality of English Language.
Comments on the Quality of English Language
  1. Lines 33-35: "Synaptic transmission is a fundamental process in normal hearing and in tinnitus, and describes communication between neurons based on chemical or electrical signals between synapses."

    This can be simpler: "Synaptic transmission is a fundamental process in normal hearing and in tinnitus, referring to the communication between neurons through chemical or electrical signals at synapses."
  2. Line 45: "top2" > "top 2"
  3. Line 45: "high degree" > "high-degree"
  4. Lines 48-50: "The analysis indicates that in SG in Tin, remodeling occurs at the cellular level under the influence of NGF and NGFR; this is associated with cell death and the formation of new cells." This can be improved to: "The analysis indicates that in the SG of Tin, remodeling occurs at the cellular level under the influence of NGF and NGFR, accompanied by both cell death and the generation of new cells."

Please note that this is only a partial review of the manuscript, and more improvements can be made in the succeeding sections that were not covered here.

Author Response

Response to Reviewer II (29. May)

Dear reviewer,

thank you very much for the valuable advice given to improve our manuscript.

Major Points

  1. The Abstract should at least cover the background, particularly the rationale/significance of the study before jumping into the methods and results. It's also crucial that the problem statement be indicated first before any discussion of methods and results.
  2. The databases (GeneCard, STRING, DAVID, Cytoscape), the specific search criteria, dataset sizes, and normalization were not discussed in detailed. How were the genes selected for each condition? Please give more context in the selection of thresholds (e.g., CS > 90th percentile). What were the filtering criteria?
  3. Statistical significance was indicated but there was no discussion of multiple testing corrections. Were the p-values corrected for multiple comparisons, particularly in GO term enrichment?
  4. The interpretations of the results seem overreaching, beyond what the data can describe. The descriptions of "xenobiotic response" and "Circadian rhythm" must be taken speculatively as these are merely associated with statistical results. I suggest using the terms "may indicate," "suggests," in connecting these interpretations with the statistical results.
  5. The Discussion and Limitations can be expanded with a discussion on the need for follow-up experimental validation as this article is purely an in-silico approach. 

Response

  1. To cover the background we add as first sentence in the abstract:

“To determine candidate key proteins involved in synaptic transmission in the thalamus in tinnitus we used bioinformatic methods by analyzing protein-protein interaction networks under different conditions of acoustic-based activity. The motor system of the thalamus (MoS) was used for comparison”.

  1.  

Specific search criteria: The analyzed genes were extracted from the GeneCards database using appropriate key terms. The GeneCards database assigns scores to genes based on various criteria, including their relevance to diseases, expression patterns, and functional annotations. The score for each condition is indicated in the Tables 1-6 (Appendix).

Search criteria STRING: Combined score > 400.

Data set sizes: Tables 1A-F (appendix) were not included in my download of the manuscript, possibly also in your download. Dataset sizes are indicated in Tables 1-6 (appendix); number of genes 31-59.

Selection of genes: From Gene cards, using suitable keywords.

As these problems also play a role in review I, we have included the following passage in the discussion to explain our approach:

3.1 Methods applied

            The identification of candidate key proteins and the analysis of molecular networks has become increasingly possible through the methods of bioinformatics and extensive freely accessible databases. The characterization of the interactions between proteins in the form of protein-protein networks can be helpful in understanding the pathomechanism of tinnitus in the individual brain centers. Proteins (nodes) that have a particularly large number of connections (degrees) to other proteins appear to play a special role in the pathophysiological mechanism of biological processes.

            The analyzed genes were extracted from the GeneCards database using appropriate key terms. GeneCards offers the possibility to identify genes that are closely related to biological processes, cellular structures and diseases, at present 3,170 genes are assigned to the keyword tinnitus. This database has not yet been used for the investigation of tinnitus. Our approach consists of the following steps: a.) Selection of genes with appropriate keywords from the GeneCards database (number of genes 20-100). b.) Analyses of the interactions of the corresponding proteins with the STRING database and the Cytoscape program. c.) Proteins with a combined score > 400 were analyzed with the DAVID database for the enrichment of GO-terms cellular components (CC) and biological processes (BP). d.) The top three high degree proteins (HDP) and the corresponding high score interaction proteins (HSIP) were selected as key proteins, whereby the 90th percentile was used as the cut-off value after analyzing the distribution of the combined score (CS) values in order to limit the study. To characterize the role of these key proteins, the GO-BP terms of these proteins were compared with that of the complete gene lists. e.) Comparison of the results with the result of a region of the thalamus that is not primarily specialized for the perception of auditory stimuli is a further indication of the validity of the results. By combining the results from different databases, which are focused on different biological contexts and indicators, reliable statements on biological processes like the synaptic transmission in tinnitus can be obtained.

  1. The DAVID database provides both the p-value for the Fischer's test and the Benjamini Hochberg value for the multiple testing corrections and determination of the false discovery rate. We have included both p-values to the corresponding figures and added in Material and method section:

“The Fishers exact test and the Benjamini-Hochberg value were used for the significance of the GO terms (p<0.01 for the gene lists and p<0.05 for the key word lists); any deviations in the value for the multiple testing corrections for key proteins are labelled separately in the legend. All GO-BP terms were included in the corresponding figures to illustrate the pathophysiological similarity of the annotations by the gene lists and the key protein lists.”

  1. We agree that the terms "xenobiotic response" and "Circadian rhythm" as well as the terms 'cocaine' and 'response to heat,' are unspecific terms, as it can be often expected with GO terms. We have tried to find a meaningful interpretation based on the knowledge of the literature. Your suggestions have been adopted. In general, the long list of Gene Ontology (GO) terms in enrichment analysis can be quite challenging due to factors like redundancy and overlap, broad vs. specific Terms, statistical significance vs. biological relevance, information overload and annotations bias.
  2. We included the following sentence in the limitations:

"It should be emphasized that this study involves the evaluation of results from biostatistical databases only; the results of our study must be experimentally validated. We are currently unable to carry out these investigations. However, the results of the study are suitable as a guide for targeted experimental investigations”

Minor Points (reviewer and response)

  1. Due to the variety of the nodes in each figure, I suggest including a figure legend to indicate what each node means for easier tracking of the readers, instead of simply describing what the node represents in the captions. -Done
  2.  A glossary should be provided to also indicate the several abbreviations used in this article, and to allow readers to track the terms easily.-Done
  3. The last paragraph of the introduction can be transferred to the Discussion or Conclusion since it covered the benefits of an in silico approach to establishing the connections between genes and proteins in these conditions. Done
  4. The manuscript's grammar and sentence structuring must be improved further. I suggest keeping sentences simple and short. Due to the complexity of the study, it may be difficult to keep track of the discussion when the sentences are too elaborate. I have some suggestions below for the improvement of Quality of English Language. Agree

Comments on the Quality of English Language

  1. Lines 33-35: "Synaptic transmission is a fundamental process in normal hearing and in tinnitus, and describes communication between neurons based on chemical or electrical signals between synapses." Done

    This can be simpler: "Synaptic transmission is a fundamental process in normal hearing and in tinnitus, referring to the communication between neurons through chemical or electrical signals at synapses." Agree
  2. Line 45: "top2" > "top 2" Done
  3. Line 45: "high degree" > "high-degree" Done
  4. Lines 48-50: "The analysis indicates that in SG in Tin, remodeling occurs at the cellular level under the influence of NGF and NGFR; this is associated with cell death and the formation of new cells." This can be improved to: "The analysis indicates that in the SG of Tin, remodeling occurs at the cellular level under the influence of NGF and NGFR, accompanied by both cell death and the generation of new cells." Agree and changed.

Please note that this is only a partial review of the manuscript, and more improvements can be made in the succeeding sections that were not covered here.

Thank you. We have revised the discussion accordingly.

Johann Gross

06.06.2025

Round 2

Reviewer 1 Report

Comments and Suggestions for Authors

Having explicitly acknowledged the key limitations of their work; the framework of the study is appropriate since it would provide a useful foundation for any following experimental investigations. Although the authors only partially addressed suggestions like cross-validation or deeper interpretation of the network metrics, the manuscript discussed these constraints and the work was sound within the scope.

Author Response

Response to reviewer I (round 2)

Dear reviewer,

Thank you very much for your comments and your help in improving our manuscript.

Kind regards,

Johann Gross

12.06.2025

Reviewer 2 Report

Comments and Suggestions for Authors

The manuscript has improved and the authors were able to address some major concerns brought up in the initial review. However, there are still some concerns that need to be addressed, and these were annotated in the manuscript. Please refer to the attached file and see the comments.

The Figure Legends are also overly long and detailed. I suggest that you illustrate the symbols and indicate what they mean in the captions, instead of describing the shapes textually. This makes the captions simpler and easier to understand. 

Comments on the Quality of English Language
  1. The capitalization of several section headers is inconsistent, such as "3.6 Limitations" while others are "3.5.2. Motor system". Please check that formatting is consistent throughout the document and adheres to the ones prescribed by the journal.
  2. Terms like "protein-protein" and "high-score" should be hyphenated.
  3. Inline references are sometimes spaced incorrectly or lack appropriate context.

Author Response

Response to reviewer II (round 2)

Dear reviewer,

Thank you very much for your comments and your help in improving our manuscript, I have tried to incorporate all your comments. The changes made are marked in blue.

Line12: (refers to your note in the ijms-thalamus-ReviewerII pdf-manuscript): Done

Line 13: Done

            -Why motor system? Response in revised manuscript ijms-3650529-round2.docx;            see line 13 and line 697

            -Rationale of our analyses? Response see line 47-51 and section 3.1: Methods

             rationale

            -Why candidate key proteins? Response see line 48-51 and line 99-100

Line 24: Done

Line 30: Done

Line 31/32: Done

Line 67: Done

Line 129: Done

Line 158: Done

Line 193: Done

Line 198: Done

Line 227: Done

Line 284:

            -Introduction HSIP; done, see line 47-51

            -CS>90th percentile cutoff; done, see line 149)

Line 315: Done (unspecific- non-specific, 5 times)

Line 405:

-When determining topological criteria of the networks, the influence of size was analysed; size has no significant effect. One example is mentioned in line 125 of the manuscript.

-Key proteins and GO terms are mainly determined by the genes or proteins itself.

-CS>90th percentile, done (see line 148)

Line 673: Done (see line 198).

Figure legends: I am sorry, with my technical means, I cannot make the suggested change.

The quality of the English language will be checked and improved by the MDPI service. Commissioned.

Section headers: Done

Hyphen and inline references: Done; I also expect correction by the MDPI Service.

Johann Gross

Berlin, 12.6.2025
